# Thermochromic aggregation-induced dual phosphorescence via temperature-dependent $sp^3$-linked donor-acceptor electronic coupling

Tao Wang[1], Zhubin Hu[2], Xiancheng Nie[1], Linkun Huang[1], Miao Hui[1], Xiang Sun [2,3 ✉] & Guoqing Zhang[1 ✉]

Aggregation-induced emission (AIE) has proven to be a viable strategy to achieve highly efficient room temperature phosphorescence (RTP) in bulk by restricting molecular motions. Here, we show that by utilizing triphenylamine (TPA) as an electronic donor that connects to an acceptor via an $sp^3$ linker, six TPA-based AIE-active RTP luminophores were obtained. Distinct dual phosphorescence bands emitting from largely localized donor and acceptor triplet emitting states could be recorded at lowered temperatures; at room temperature, only a merged RTP band is present. Theoretical investigations reveal that the two temperature-dependent phosphorescence bands both originate from local/global minima from the lowest triplet excited state ($T_1$). The reported molecular construct serves as an intermediary case between a fully conjugated donor-acceptor system and a donor/acceptor binary mix, which may provide important clues on the design and control of high-freedom molecular systems with complex excited-state dynamics.

[1] Hefei National Laboratory for Physical Science at the Microscale, University of Science and Technology of China, Hefei, China. [2] Division of Arts and Science, NYU-ECNU Center for Computational Chemistry, NYU Shanghai, Shanghai, China. [3] Department of Chemistry, New York University, New York, NY, USA. ✉email: xiang.sun@nyu.edu; gzhang@ustc.edu.cn

Suppressing nonradiative decays (e.g., intramolecular motions) and promoting the intersystem crossing rate (ISC) are crucial to realize organic room-temperature phosphorescence (RTP) intense enough for practical applications[1,2]. Up till now, several RTP design principles have been proposed, such as introducing bulky hindrance groups[3,4] and spiro linkers[5], constructing aggregates[6–8], polymers[9,10] and self-assemblies[11–13], host–guest[14–16] and bonding[17–19] interactions, as well as tuning charge-transfer (CT) states[20,21]. Recently, we put forward a new RTP design principle based on an $sp^3$-linker connected donor–acceptor dyad, where an angled intramolecular CT state is believed to enhance spin–orbit coupling due to electron circular motion during transition[22,23]. For example, a proton-activated off–on RTP molecular probe could be developed based on this principle[22]. However, most RTP emitters still require stringent environmental factors, such as low temperature, solid matrix assistance, and/or oxygen exclusion, which can be rather cumbersome for practical applications. Since 2001 (ref. [24]), aggregation-induced emission (AIE) luminophores have been a research hotspot because of their enormous potentials in solid-state display[25–27], bio-probes[28–30], circularly polarized luminescence[31,32], and on–off sensors[33], with an advantage being emitting in their own microenvironments. Therefore, a universal design method for the development of RTP materials may be built upon less environmental-sensitive AIE core structures with RTP's increasing use in chemical sensing[34,35], data storage[36,37], bioimaging[38,39], and OLEDs[40,41]. Herein, we present an RTP design strategy of molecular solids by combining the concept of AIE and the donor–$sp^3$ linker–acceptor dyad molecular motif and demonstrate that such a design yields an unexpected phenomenon of thermochromic phosphorescence (TCP). We also show that the temperature-dependent emission is largely dictated by

how strongly the two largely locally excited triplet emitting states (i.e., donor triplet $^3LE_D$ and acceptor triplet $^3LE_A$) associate (Fig. 1a). The TPA moiety was selected as an AIE activator, as well as an electron donor with the acceptor varying in chemical structures. In Fig. 1b, five TPA-functionalized AIE-active RTP molecules were synthesized via simple Suzuki–Miyaura or Ullmann-type coupling reactions (Supplementary Fig. 1), and the AIE properties of TPA1-5 were investigated.

## Results

**AIE characterization.** In solution, absorption spectra conducted in THF (Fig. 2a) are consistent with calculations by the time-dependent density functional theory (TD-DFT) method with the optimally tuning range-separated functional (LC-ωPBE*) and the TZVP basis set[42,43]. According to the theoretical results (Supplementary Fig. 2 and Supplementary Table 1), the reddest absorption band (>280 nm) of TPA1-2 is mainly contributed by the second and the third singlet excited states ($S_2$ and $S_3$), while that of TPA3–5 is dominated by $S_3$ and the fourth singlet excited state ($S_4$). The hole–electron distribution analysis performed by Multiwfn 3.7 (dev) program[44] indicates that lower electronic transitions of all TPAs are mainly contributed by local excitations (Fig. 2a, right), and the lowest singlet excited state ($S_1$) is a dark state with a strong CT character, consistent with weak absorption trailing into the near-UV region. The fact that the absorption maxima in various solvents exhibit almost no difference also indicates the main bands belong to transitions to localized excited states (e.g., $S_2$ and $S_3$, Supplementary Fig. 3). Therefore, it can be inferred that TPAs are non-emissive due to a combination of excited-state energy dissipation via intramolecular rotation (propeller-shaped TPA moiety) and a lowest forbidden CT pathway. With increasing water fraction, a typical AIE process

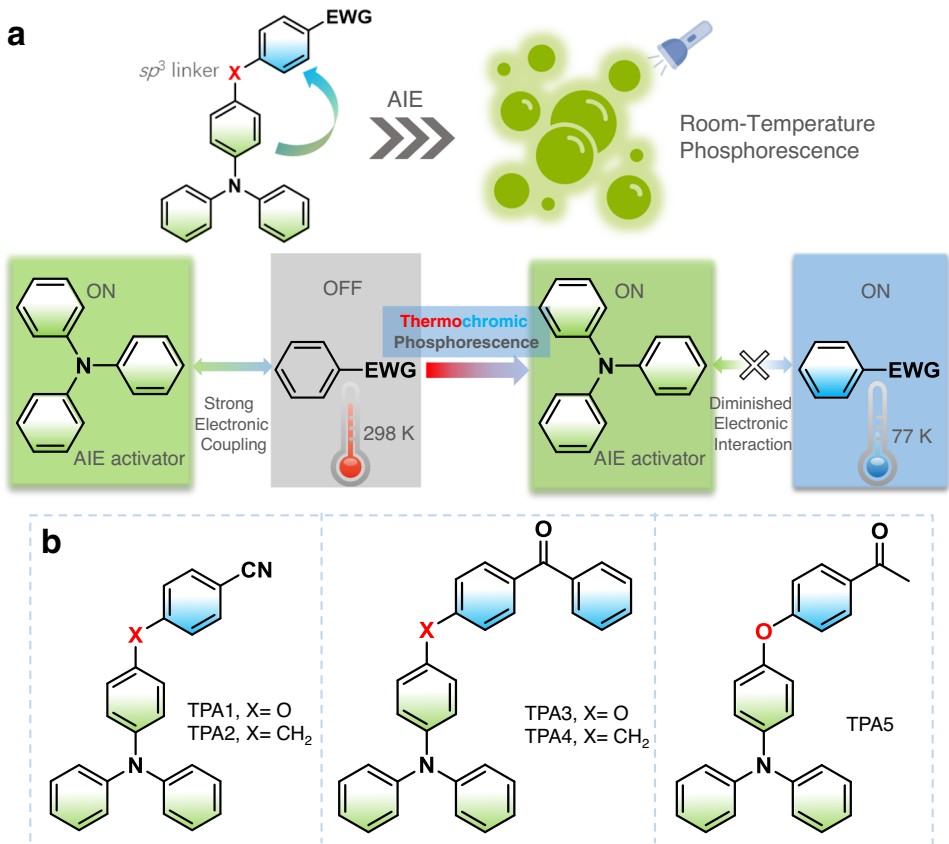

**Fig. 1 AIE-active dual-phosphorescence molecular design. a** Design concept of AIE-gen donor–$sp^3$ linker–acceptor thermochromic dual phosphorescent dyad via temperature-dependent coupling between two emissive triplet excited states. **b** Related chemical structures. EWG = electron-withdrawing group.

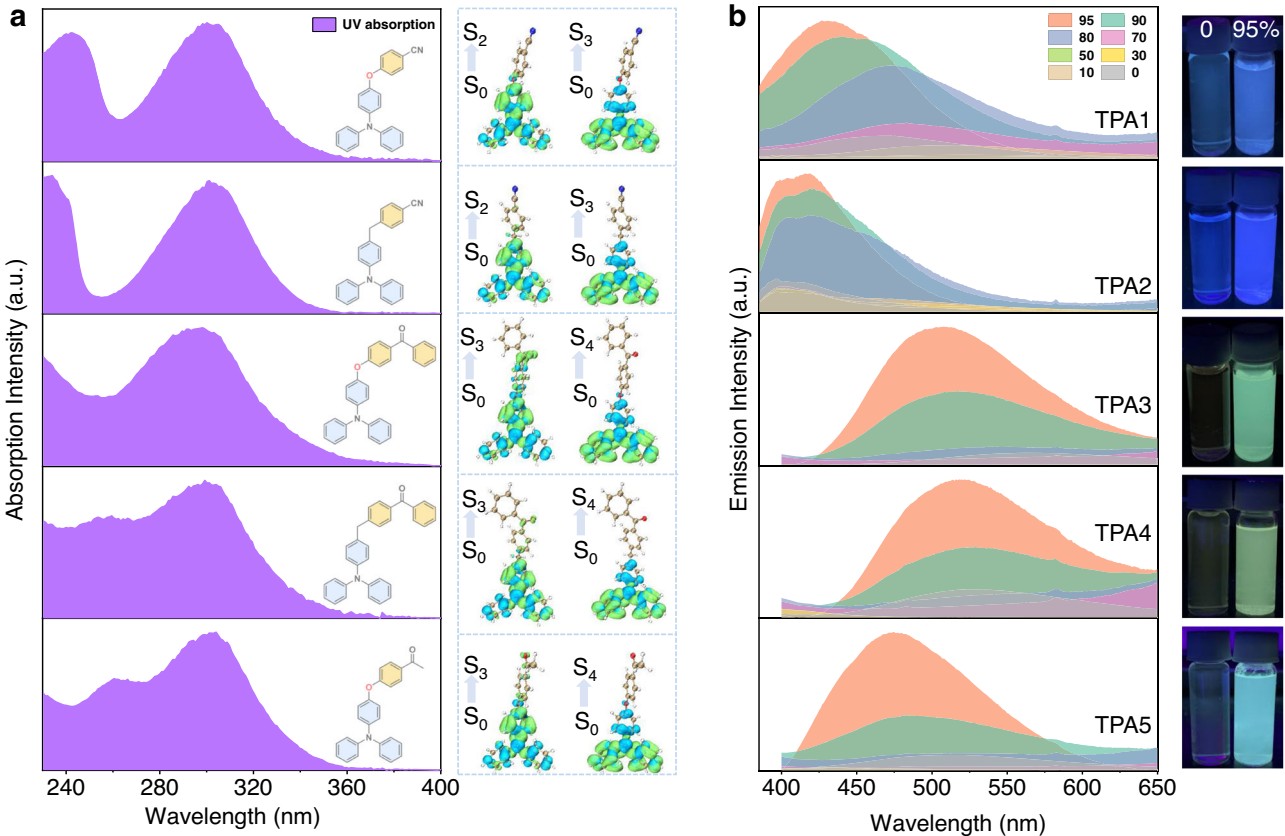

**Fig. 2 AIE investigations. a** Normalized absorption spectra (left) of TPA1-5 in optically dilute THF solutions and corresponding hole (blue)–electron (green) distributions (right) calculated based on the TD-DFT method. **b** Steady-state emission spectra of TPA1-5 in a bicomponent solution mixture showing the AIE process with different THF/water ratios (0–95%, v/v) and related AIE photos under a hand-held UV lamp (water fraction: 0 and 95%, TPA concentration: $2.0 \times 10^{-3}$ mol/L).

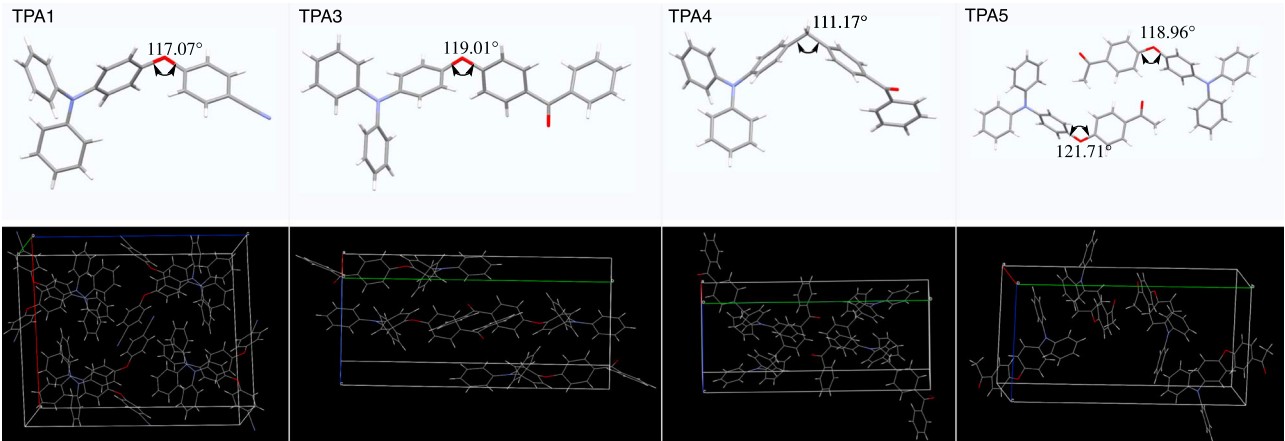

**Fig. 3 Single-crystal XRD analysis.** Stick models (top) of TPA1 and TPA3–5 obtained from single-crystal XRD measurements and packing diagrams shown in wireframe model (bottom), indicating no obvious strong π–π intermolecular interactions in the solid state.

(Fig. 2b) can be observed due to the restriction of intramolecular motions and possibly newly emerged, lowest emissive states as aggregates. It has to be noted that the observed emission maximum shift mainly results from the change of mixture solvent polarity[45]. Moreover, all TPA aggregates in THF/water (5/95, v/v) contain long-lived emissions, and the afterglow can even be observed for certain (e.g., TPA1) aggregates (Supplementary Fig. 4). The scanning electron microscopy (SEM) images (Supplementary Fig. 5) show that some of these aggregates are in fact

ordered and form nanocrystals (TPA1, TPA2, and TPA5); the increased particle sizes of TPA3 and TPA4 may originate from reduced intermolecular interactions given that they exhibit high molecular complexity.

To assist further understanding of their AIE behaviors, single-crystal measurements of TPA1 and TPA3–5 were conducted to investigate the origin of the emission, except for TPA2 which always generates polycrystals. As shown in Fig. 3, the donor and acceptor moieties of TPAs adopt a highly twisted conformation

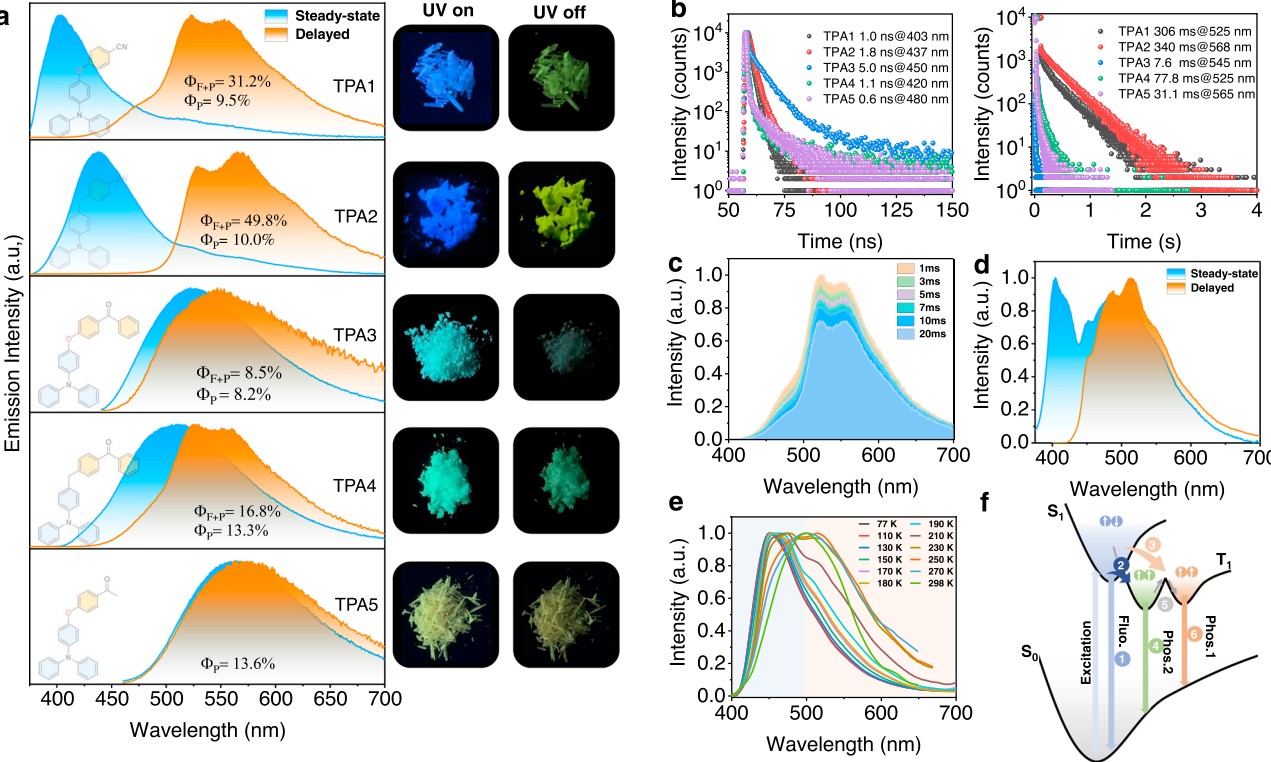

**Fig. 4 Photophysical investigations for TPA1-5 molecular solids. a** Normalized emission spectra of TPA1-5 in air at room temperature (excitation: 365 nm for TPA1, TPA2, and TPA4; 430 nm for TPA3; and 450 nm for TPA5) and corresponding photos showing TPA1-5 excited by 365-nm UV light. **b** Time-resolved decay profiles of TPA1-5 in air at room temperature (right: fluorescence; left: RTP). **c** Time-resolved emission spectra of TPA1. **d** Normalized emission spectra of TPA1 at 77 K. **e** Normalized phosphorescence emission spectra of TPA1 dissolved in PMMA film (excitation: 375 nm; concentration: 3%, w/w) at different temperatures. **f** Schematic illustration of the ternary emission process; process 1: fluorescence; process 2: intersystem crossing at 77 and 298 K; process 3: favored intersystem crossing at 298 K; process 4 and 6: phosphorescence; and process 5: thermally activated conformation transformation.

due to the separation by a $sp^3$ linker (O or $CH_2$), resulting in a twist angle within a range of ~110–120°. As expected, the combined effect of a propeller-shaped TPA donor and an additional twist exerted by the $sp^3$ linker is apparent: it makes π–π stacking essentially nonexistent, while suppressing various molecular motions as aggregates, which is a typical AIE mechanism.

**Photophysical properties in the aggregation state**. These collected aggregates were then investigated by both steady-state and delayed luminescence spectroscopy. The optimal excitation wavelengths were assessed from solid-state UV–vis absorption and excitation spectra (Supplementary Fig. 6). For TPA1 and TPA2, the steady-state emission exhibits a main AIE-fluorescence band ($\lambda_{em}$ = 403 and 437 nm, respectively, Fig. 4a) with nanosecond lifetimes (Fig. 4b, left). A low-energy band over 500 nm with a lifetime of >300 ms (Fig. 4b, right) is presumably due to the AIE-RTP. When the delayed spectrum of TPA1 was examined, we observed a high-energy broad shoulder (~485 nm), that is, >1600 $cm^{-1}$ above from lowest triplet excited-state emission ($T_1$). To shed light on the origin of this shoulder peak, time-resolved emission spectra were collected (Fig. 4c), where the shoulder peak intensity decays faster vs. that of the main RTP peak at room temperature (Supplementary Figs. 7, 8a). At 77 K, the steady-state and delayed emissions show a tremendous increase in the shoulder band (Fig. 4d), which perhaps indicates a second emissive triplet state[2]. For delayed emissions, the relative intensity ratio between the shoulder band and the main peak maintains

constant (Supplementary Fig. 9). This temperature-dependent phosphorescence decay kinetics suggests that the second emissive triplet excited state is feeding the lowest $T_1$ at room temperature, whereas the communication is cut off at 77 K.

To exclude the influence of ground-state aggregation and trace impurity contamination (which can be significant in molecular solids), the phosphorescence spectra ($\Delta t$ = 1 ms) of single-crystal purity TPA1 dissolved in PMMA were also recorded at various temperatures, where a consistently blueshifted trend could be noted with the decreasing temperature (Fig. 4e), further indicative of such a second triplet state being separated from the lowest $T_1$. Visually, a color change from green to sky blue in the afterglow could be noted. A schematic illustration for explaining this ternary emission (fluorescence and dual phosphoresce) is presented in Fig. 4f. For highest twisted molecules with many vibrational and rotational freedoms like these TPA derivatives, we anticipate a very rough potential energy surface in the $T_1$ state: more than one local minima can therefore become the emitting states. It is reasonable to assume that a molecular geometry further away from the equilibrium position gives off emission at a longer wavelength, which is designated as $T_1^L$; the other emitting state closer to equilibrium position is denoted as $T_1^H$ (L/H stands for low/high). The temperature dependency can therefore be interpreted as: (1) a higher temperature produces hotter excitons that may prefer ISC favorable for relaxation to the $T_1^L$ site and vice versa; (2) communications among these emitting states at local minima by thermal motions (e.g., vibrations) may be cut off at frigid temperatures, so that more distinct separation in spectrum could be revealed.

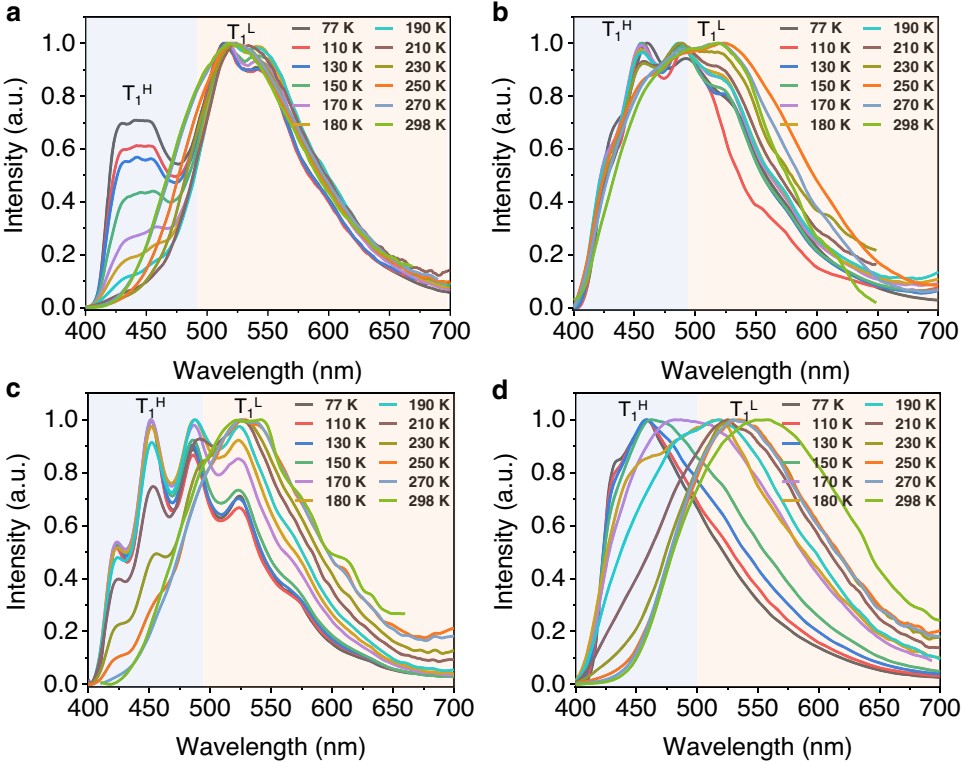

**Fig. 5 Temperature-dependent time-resolved emissions of PMMA films for showing dual phosphorescence.** Normalized phosphorescence emission spectra of TPA2 (**a**), TPA3 (**b**), TPA4 (**c**), and TPA5 (**d**) dissolved in PMMA film ($\Delta t = 1$ ms; excitation: 375 nm; and concentration: 3%, w/w) at different temperatures showing the two triplet-state emissions.

For TPA2 aggregates, although little to no $T_1^H$ emission band is presented at room temperature (Fig. 4a and Supplementary Fig. 10), a new peak shows up in the low-temperature phosphorescence spectrum (77 K, Supplementary Fig. 11) at 455 nm in addition to the main band ($\lambda = 525$ nm), indicating a higher energy barrier between the two emitting states when the $CH_2$ linker replaces O at room temperature. Similarly, when TPA2 was molecularly dissolved in the PMMA film, a much stronger high-energy shoulder peak ($\lambda = 440$ nm) gradually emerged (Fig. 5a), compared to the aggregated state. The experimental data above appear to indicate that two $T_1$ states communicate at room temperature both intermolecularly and intramolecularly, since the aggregates tend to give a much weaker $T_1^H$ phosphorescence band irrespectively of the temperature, compared to discrete molecules in PMMA.

At this stage, we speculate that the $T_1^L$ state is a TPA-localized triplet state with $T_1^H$ being an acceptor (cyanobenzene) centric triplet one. To verify the hypothesis, two important experiments were conducted: (1) TPA3 to TPA5 with an aromatic ketone acceptor were prepared. It is well-known that benzophenone exhibits a prominent $^3n-\pi^*$ state[46,47] with a characteristic vibrational progression in the phosphorescence spectrum separated by ~1300–1600 cm$^{-1}$. Indeed, such spectroscopic patterns from the $T_1^H$ state characteristic of benzophenone $^3n-\pi^*$ were unambiguously obtained (Fig. 5b, c), when the temperature is reduced to or <200 K for the two samples dissolved in PMMA films (particularly for the $CH_2$-linked TPA4, where no oxygen conjugation interferes with the vibration), further revealing that phosphorescence emissions consist of two emissive triplet states from benzophenone and TPA subunits, respectively. It has to be noted that <150 K, the $T_1^L$ state appears largely suppressed, again suggesting preferential ISC relaxations at different temperatures (Fig. 4f).

For TPA5, the *para*-oxygen substituted acetophenone acceptor is expected to exhibit a mixed $^3\pi-\pi^*$ and $^3n-\pi^*$ band for $T_1^H$ in PMMA at 77 K ($\lambda_{em} = 455$ nm; Fig. 5d). Temperature-dependent excitation spectra (Supplementary Fig. 12) of TPA1-5 dissolved in PMMA films also suggest the existence of two different emissive triplet states. In the aggregated states, all three samples (TPA3–5) exhibit a broad RTP emission dominated by $T_1^L$ (Fig. 4a) and much shorter RTP lifetimes (Fig. 4b and Supplementary Table 2) due to strong electronic coupling between the TPA-localized $^3\pi-\pi^*$ and the ketone-localized $^3n-\pi^*$ state; such coupling is also present, albeit weak, at 77 K based on the phosphorescence spectra (Supplementary Figs. 13–15). Notably, due to the introduction of the carbonyl group in TPA3–5, the fluorescence emission almost totally turns into RTP because of a promoted ISC process, and the fluorescence emission can only be measured before 480 nm (Fig. 4b, left). A binary molecular mixture between TPA and the benzophenone precursor at 1:1 ratio was used to repeat the same experiment (Supplementary Fig. 16), where it was found that the 1:1 physical mixture generates RTP bands belonging to the two molecules irrespective of the temperature. The results not only suggest that the two RTP bands do originate from the donor and acceptor, respectively, but also point to the importance of the $sp^3$ chemical linker: cutting off communications at low temperature but not high temperature, something completely different from blending.

**Computational investigation on the origin of dual phosphorescence.** To further back the proposed $T_1^H$–$T_1^L$ dual-phosphorescence states model, we first selected TPA1 as example to carry out a series of theoretical calculations via TD-DFT at the LC-ωPBE*/TZVP level in the gas state, which mimics the state of monomers in PMMA. The calculated energy diagram of the vertical excitation and spin–orbit coupling constants reveals multiple ISC

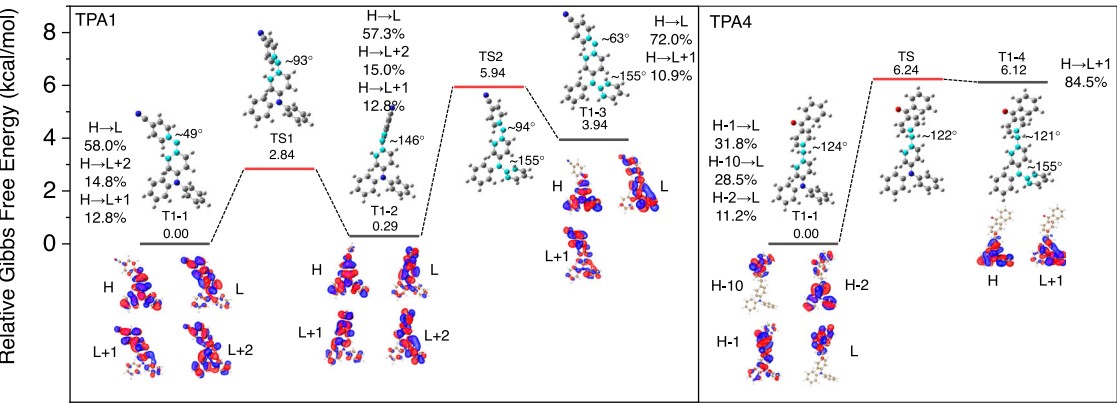

**Fig. 6 Theoretical computations on the conformations and molecular orbital transitions for dual phosphorescence.** Calculated relative Gibbs free energies and global/local-minima geometries (T1–1, T1–2, T1–3, and T1–4) of $T_1$ state, the related transition states (TS) between these geometries, the optimal geometries with dihedrals of highlighted atoms, and main molecular orbital transition contributions for $T_1$ emissions (>10%; H HOMO, L LUMO).

channels from $S_1$ to a higher triplet state during RTP generation (Supplementary Fig. 17), pointing to the possibility of preferential ISC channels depending on temperature. Moreover, the vertical emission calculations demonstrate that (1) the $S_1$ state is almost degenerate with the acceptor $T_2$ state, which boosts the ISC process, and (2) the energy gap between $T_1$ and $T_2$ states is relatively large (Supplementary Fig. 18). However, according to the energy gap law for internal conversion[48]: $\log(k_{IC}) \approx 12 - 2\Delta\nu$, where $k_{IC}$ ($s^{-1}$) is the internal conversion rate and $\Delta\nu$ ($\mu m^{-1}$) is the wavenumber difference between $T_1$ and $T_2$, the calculated $k_{IC}$ for TPA1 is $\sim 3.95 \times 10^{-10}\ s^{-1}$, which indicates that triplet excitons can only emit as photons from $T_1$.

To shed light on the origin of dual phosphorescence, we further conducted relative Gibbs free energies and the three global/local-minima geometries (T1–1 to T1–3) of $T_1$-state TPA1 (Fig. 6). As Fig. 6 shows, there are different TPA1 conformations showing different $T_1$ states, which exhibit $T_1$ emissions contributed by donor and acceptor. Due to the existence of energy barriers (TS1 and TS2), the conformation transformation should be decided by transition energy, which can be compensated at room temperature while inhibited at low temperature, such as 77 K. $T_1^H$ and $T_1^L$ are not completely localized on donor and acceptor of TPA1 perhaps because cyanobenzene is a type of extremely weak emitter, which is consistent with spectroscopic evidence that TPA1 $T_1^L$ emission is always dominant. Therefore, we further conducted the calculations of TPA4, where benzophenone is a better emitter than cyanobenzene. As anticipated, two $T_1$ species are localized on TPA and benzophenone subunits, perfectly corresponding with the experimental results (Fig. 5c and Supplementary Fig. 14). Therefore, we can conclude that the dual phosphorescence mainly originates from $T_1$ states of TPA derivatives in specific conformations, lining up with the dual-phosphorescence model in Fig. 4f.

**Application explorations.** Finally, to showcase the application value of the AIE thermochromic dual phosphorescence molecules, we synthesized TPA6 with a TPA donor and a pyridine acceptor (Fig. 7a). The rationale is that the electron-withdrawing pyridyl group is smaller than cyanobenzene, which should yield an even more separated $T_1^H$ and $T_1^L$ energy gap to make the visual phosphorescence color change more dramatic and spectroscopically more resolvable. Meanwhile, the lone pair electron in the pyridine moiety is likely to make molecular stacking even more difficult in the solid state, so that no PMMA matrix is needed for the TPA6-based phosphorescence sensing module.

Figure 7b shows that the TPA6 solid exhibits a fluorescence emission at 425 nm ($\tau = 0.7$ ns). Meanwhile, an obvious afterglow can be observed by the naked eye and an RTP emission band can be collected at 550 nm with a lifetime of 83 ms in air at room temperature. As the temperature decreases, a new phosphorescence band ($T_1^H$) appears with an emission maximum at 455 nm in the steady-state emission spectrum (Fig. 7c), which can be used as a ratiometric phosphorescence/fluorescence sensing scheme (Fig. 7d) demonstrated in many previous reports[49,50]. However, here we show for the first time that a delayed spectrum can further become sensitive to an external stimulus and responds to temperature change dramatically (Fig. 7e, f), which could be potentially used as the thermometer for indicating extremely low-temperature environment.

## Discussion

In summary, by introducing a TPA-based AIE-gen into the donor–$sp^3$ linker–acceptor structure, several ternary emissive AIE-active RTP molecules with prompt fluorescence and dual phosphorescence were obtained. The phenomenon is explained by the presence of a TPA-localized $T_1^L$ state, which couples to a higher acceptor $T_1^H$ state at room temperature, but cuts the electronic coupling off at lowered temperatures. Therefore, a universal principle for designing a dual-phosphorescence thermochromic material can be deduced and demonstrated. The current strategy benefits design for single-component, dual-phosphorescence-based molecular probes. More importantly, we expect many more structurally different molecular can be used in this strategy with a possibility of generating more than two phosphorescence bands to obtain kinetically more complex molecular systems.

## Methods

**Synthesis**. TPA1-6 were synthesized by Suzuki–Miyaura and Ullmann-type coupling reactions. Further detailed information on synthetic procedures is provided as Supporting Information.

**Measurements**. $^1H$ NMR (400 MHz) spectra were collected on Bruker AV400 NMR spectrometer and operated in the Fourier transform mode at 298 K. The related chemical shifts were reported as values in p.p.m. relative to tetra-methylsilane ($\delta = 0$) in deuterated solvents. High-resolution mass spectra were conducted on an Atouflex speed mass spectrometer using the electrospray ionization mode. Single-crystal data were collected from a XtaLAB AFC12 (RINC): Kappa single diffractometer. The crystal was kept at 293 K during data collection. Using Olex2 (ref. [51]), the structure was solved with the ShelXT structure solution program using intrinsic phasing[52], and refined with the ShelXL refinement package using least squares. The morphologies of the aggregates in THF/$H_2O$ (5/95, v/v)

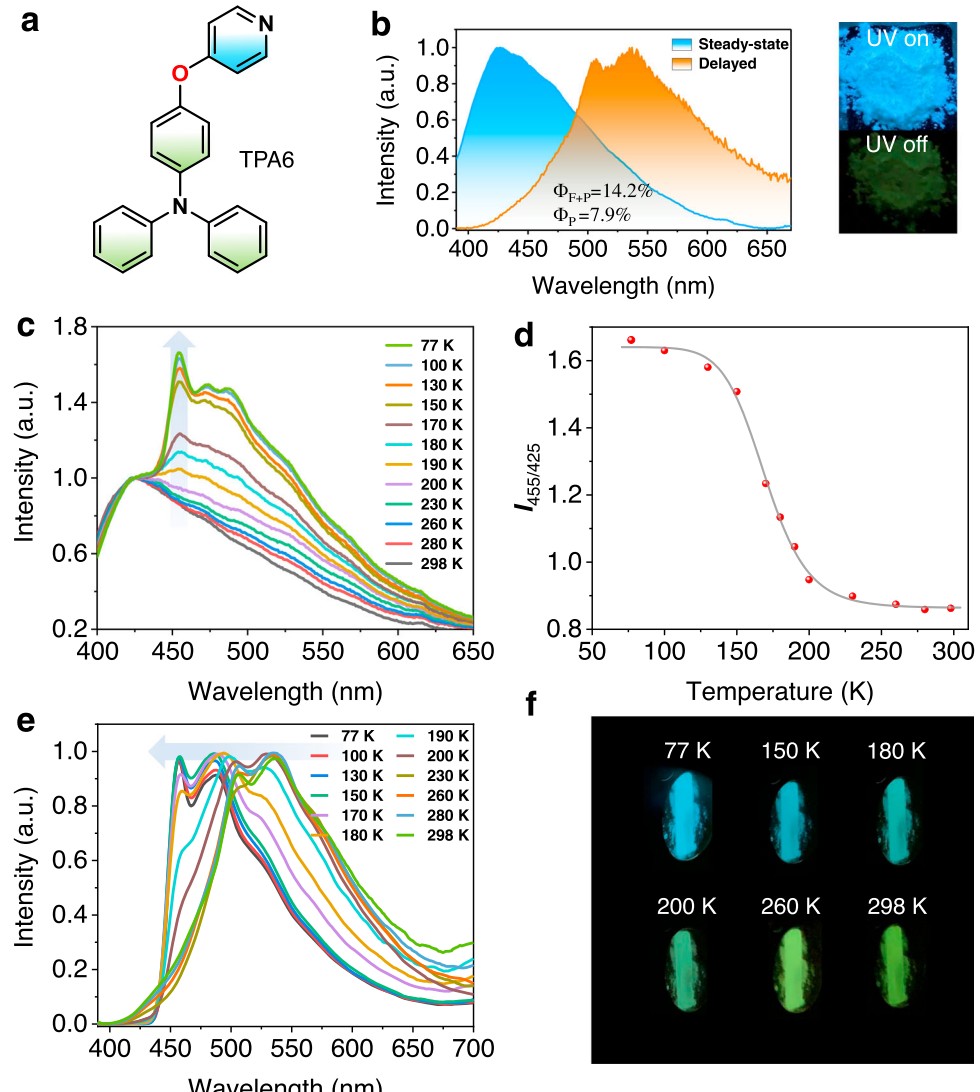

**Fig. 7 Photophysical properties of TPA6. a** Chemical structure of TPA6. **b** Normalized steady-state and RTP emission ($\Delta t = 3$ ms) spectra of TPA6 in air at room temperature; inset: photo showing TPA6 solid excited by 365-nm UV light. **c** Relative steady-state emission spectra of TPA6 at different temperatures. **d** Relationship between the emission intensity ratio of 455 and 425 nm ($I_{455}/I_{425}$) and temperature. **e** Normalized temperature-dependent phosphorescence emission spectra of TPA6. **f** Image showing delayed emission (>50 ms) color change at different temperatures.

were investigated via field-emission SEM (JEOL JSM-6700F, 10 kV). UV–vis spectra were measured in THF on Agilent Cary 60 UV–vis spectrometer ranging from 190 to 1100 nm and data processed on Origin 2020. Steady-state and time-resolved photoluminescence emission spectra were conducted on a FluoroMax-4 spectrofluorometer (Horiba Scientific) and analyzed by Origin 2020 software. Absolute quantum yields were determined using an integrating sphere. Emission lifetime decay profiles were collected on Horiba Ultima time-resolved fluorescence spectrometer with 1 MHz lasers, and the related data were analyzed with DataStation v6.6 (Horiba Scientific).

## Data availability

All relevant data that support the findings are available within this article and Supporting Information and are also available from authors upon reasonable request. The crystallographic data for TPA1, TPA3, TPA4, and TPA5, have been deposited in the Cambridge Crystallographic Data Center (CCDC; https://www.ccdc.cam.ac.uk/structures/) under accession numbers CCDC: 2022517, 2022518, 2022520, and 2022519.

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

## Acknowledgements

We would like to thank the National Key R&D Program of China (2017YFA0303500 to G.Z.) and the National Natural Science Foundation of China (21975238 to G.Z. and 21903054 to X.S.). We are extremely grateful for the invaluable discussion with Prof. Yi Luo from USTC.

## Author contributions

T.W. and G.Z. conceived and designed the project. T.W. synthesized all the related target compounds. T.W., X.N., L.H., and H.M. performed the relevant photophysical measurements. Z.H. and X.S. conducted the theoretical calculations. T.W. and G.Z. wrote and revised the manuscript. All authors discussed the results and commented on the manuscript.

## Competing interests

The authors declare no competing interests.
