## [Peer Review File · Nature Communications]

REVIEWER COMMENTS

Reviewer #1 (Remarks to the Author):

Wang et al. designed several ternary emissive AIE-active RTP molecules with prompt fluorescence and dual phosphorescence by introducing TPA-based AIE-gen into the donor-sp³ linker-acceptor structure. In general, the procedure and the results of the work were presented in this paper clearly, while there remain some details to be optimized. I don't recommend this manuscript to be published in Nat. Commun..

1. The absorption and excitation spectra of all the compounds in the aggregation state should be provided.
2. In Figure 4, the emission was excited at 365, 430 even 450 nm light. However, those compounds didn't show evident absorption when the wavelength larger than 360 nm. Why those excitation wavelengths were chosen? What will happen if excited those compounds at a corresponding maximum absorption wavelength?
3. The explanation about the multi-emission is unreasonable. First, according to the author's data, the k_P of "T1" and "Tn" was about 100 or 101 s⁻¹. However, the k_{IC} was much larger than it (generally larger than 10¹² s⁻¹). The T₂→S₀ transition couldn't compete with the internal conversion from T₂→T₁, which means the bluer phosphorescence shouldn't not original from the T_n state. Second, in line 137 the author said "At this stage, we speculate that the T₁ state is a TPA-localized triplet state with T_n being an acceptor (cyanobenzene) centric triplet one, given that the sp³ linker renders the two moieties more spatially separated vs. an sp² counterpart and thus less electronically coupled at low temperature when vibrations are inhibited.", the vibration did will be inhibited by low temperature, however, according to the theory calculation, the excited electron in T₂ state was mainly located at TPA segment, same as the T₁ state. Thereby, the temperature won't influence the IC between T₂ and T₁. Besides, the IC rate was considered to be constant for temperature. The explanation is unpersuasive.
4. Researches using anti-Kasha's rule to obtain and demonstrate dual phosphorescence have been reported. In 2017, Tang group (Nat. Commun. 2017, 8, 416) reported the dual RTP phenomenon which arise from the low- and high-lying triplet states.
5. In Figure 1, authors should provide the full name of EWG.
6. We suggest the author to use X-ray diffraction spectra to investigate the microstructure of the aggregates in THF/H₂O mixtures.
7. On Page 9 of the manuscript, line NO.139, the authors mentioned that "an sp² counterpart and thus less electronically coupled at low temperature when vibrations are inhibited" The authors should provide the photophysical data of counterpart molecules.

Reviewer #2 (Remarks to the Author):

In this manuscript, Zhang et al. reported that donor-sp³ linker-acceptor structures can generate a fascinating dual phosphorescence, which is proved tunable via modulation of temperature as a result of strong or weak electronic coupling. Furthermore, through rationally decrease the electron-

withdrawing ability of acceptor group, the authors made the visual phosphorescence color change more dramatic. This finding offered a vital clue on modulating complex excited-state dynamics in molecular systems, indicating a universal principle for designing dual-phosphorescence thermochromic materials. The elaborate experimental data and theoretical calculations signify the excellent quality of this work. Considering its novelty and foreseeable wide-ranging influence in many related fields, therefore, I strongly recommend its publication in Nature Communications, after the authors address the following concerns.

1. Figure 1 shows that all five TPA compounds exhibit local excitation, while the authors stated that the S1 state is a dark state with the CT character. If so, why do the emission spectra show CT emission character? The authors should give more explanation.
2. From Figure 6, all transitions for T1 and T2 include the CT character. How do the authors confirm the radiative decays are from donor and acceptor, respectively?
3. In caption of Figure 6, a letter “d” must be lost.
4. For TPA6, it is interesting that the T2 band increases more obviously than T1 with temperature decreasing. I suggest the authors to give some detailed explanations.
5. According to UK English, “behavior” and “color” are encouraged to be replaced as “behaviour” and “colour”, respectively.

Reviewer #3 (Remarks to the Author):

The manuscript by Zhang et al. described a series of purely organic AIE-active luminophores with efficient dual phosphorescence. By utilizing an sp³ linker as the spacer between the donor and acceptor to break the spatial overlap between HOMO and LUMO, efficient ISC process can be realized. The photophysics were fully understood by DFT calculations and temperature-dependent PL analysis. This work shows “donor-sp³ linker-acceptor” should have the potential to become a new generation of purely organic RTP building block and platform, which will be inspirational and of great significance for the RTP community. Therefore, I would like to recommend its timely publication in Nat. Commun. after the following minor revisions.

1. As shown in the MS, all these compounds consist of donor and acceptor subunits. Figure 1a shows the major UV absorption peaks around 300 nm, mainly ascribed to local excitation of TPA. Are these peaks influenced by solvents? The UV-vis absorption spectra in different solvents should be collected.
2. For the dual phosphorescence phenomenon, the author assigned them as emissions from the donor and acceptor, respectively. How does the temperature influence the energy transfer process between the two different triplet states?
3. To better understand the origin of dual phosphorescence, NTO transition calculations should be carried out.

4. Given that in the aggregated state for TPA2-5 there is almost no RTP from a higher triplet state, do intermolecular interactions significantly influence the dual phosphorescence?
5. For Figure 5 caption, "showing two triplet-state emissions" should be added for a better understanding of these spectra.
6. A minor typo: in figure 7, the digits of I455/425 should be subscript.

Replies to Reviewers' Comment for Manuscript NCOMMS-20-32351

Responses to the Comments and Suggestions of Reviewer 1

General Comment: Wang et al. designed several ternary emissive AIE-active RTP molecules with prompt fluorescence and dual phosphorescence by introducing TPA-based AIE-gen into the donor-sp³ linker-acceptor structure. In general, the procedure and the results of the work were presented in this paper clearly, while there remain some details to be optimized. I don't recommend this manuscript to be published in Nat. Commun.

Our reply: We appreciate the valuable comments from the reviewer on our work. As can be seen, the comments are vital improvement of the quality of the revised manuscript. Our point-to-point response is presented below. We hope that our responses, along with optimized details, can clear the issues raised by the reviewer.

1. The absorption and excitation spectra of all the compounds in the aggregation state should be provided.

Our reply: We thank the reviewer for this valuable advice. The related absorption and excitation spectra in the aggregation state have been measured and presented in Supplementary Figure 4. Different from UV-vis absorptions of solution (Figure 1a), the solid-state absorptions (Supplementary Figure 4) show a weak CT transition (>400 nm), which is not uncommon due to preferred molecular geometry (for intramolecular CT) in the solid state, new intermolecular excitation modes, or a combination of both. The solid-state excitation spectra show an evident emission band beyond 350 nm for **TPA1-6**, especially **TPA3** and **TPA5**, whose excitation maxima is ~430 nm and 450 nm.

Supplementary Figure 4. (a) Solid-state UV-vis absorption and (b) excitation spectra of **TPA1-6**.

2. In Figure 4, the emission was excited at 365, 430 even 450 nm light. However, those compounds didn't show evident absorption when the wavelength larger than

360 nm. Why those excitation wavelengths were chosen? What will happen if excited those compounds at a corresponding maximum absorption wavelength?

Our reply: We thank the reviewer for raising this question, which is in fact a continuation from the first question. The excitation wavelengths for solid-state emissions were chosen according to correspondent solid-state excitation spectra, not the absorption spectra presented in Figure 1a. As shown in Figure 4b, **TPA1**, **TPA2**, **TPA4**, and **TPA6** exhibit optimal excitation wavelength around 350-400 nm. After comparing with emission spectra, we finally chose 365 nm as the best excitation wavelength. For **TPA3** and **TPA5**, we can clearly see the optimal excitation wavelengths are ~430 and 450 nm, respectively.

3. The explanation about the multi-emission is unreasonable. First, according to the author's data, the k_P of "T1" and "Tn" was about 100 or 101 s⁻¹. However, the k_{IC} was much larger than it (generally larger than 10¹² s⁻¹). The T₂→S₀ transition couldn't compete with the internal conversion from T₂→T₁, which means the bluer phosphorescence shouldn't not original from the T_n state. Second, in line 137 the author said "At this stage, we speculate that the T₁ state is a TPA-localized triplet state with T_n being an acceptor (cyanobenzene) centric triplet one, given that the sp³ linker renders the two moieties more spatially separated vs. an sp² counterpart and thus less electronically coupled at low temperature when vibrations are inhibited.", the vibration did will be inhibited by low temperature, however, according to the theory calculation, the excited electron in T₂ state was mainly located at TPA segment, same as the T₁ state. Thereby, the temperature won't influence the IC between T₂ and T₁. Besides, the IC rate was considered to be constant for temperature. The explanation is unpersuasive.

Our reply: This is where we express our biggest appreciation for the question the reviewer raised. In order to address the discrepancy between the experimental results and a persuasive explanation, we performed calculations at a higher resolution and try to examine whether such phenomenon could be related to conformational changes, i.e., the possibility of more than one emitting states on the T₁ potential energy surface.

Indeed, the calculations for two representative molecules, do suggest that multiple T₁ emitting states are possible given their very high degree of conformational freedom. We have substantially revised the manuscript to work in the new interpretation where the related description has been presented in the **Photophysical properties in the aggregation state** and the **Computational investigation on the origin of dual phosphorescence** sections.

The current text reads in the **Photophysical properties in the aggregation state** section: "For highest twisted molecules with many vibrational and rotational freedoms like these **TPA** derivatives, we anticipate a very rough potential energy surface (PES) in the T₁ state: more than one local minima can therefore become the emitting states.

It is reasonable to assume that a molecular geometry further away from the equilibrium position gives off emission at a longer wavelength, which is designated as T_1^L ; the other emitting state closer to equilibrium position is denoted as T_1^H (L/H stands for low/high). The temperature dependency can therefore be interpreted as: 1) a higher temperature produces hotter excitons that may prefer intersystem crossing (ISC) favourable for relaxation to the T_1^L site and vice versa; 2) communications among these emitting states at local minima by thermal motions (e.g., vibrations) may be cut off at frigid temperatures, so that more distinct separation in spectrum could be revealed.”

And in the **Computational investigation on the origin of dual phosphorescence** section: “However, according to the energy gap law for internal conversion⁴⁸: $\log(k_{IC}) \approx 12-2\Delta\nu$, where k_{IC} (s^{-1}) is the internal conversion rate and $\Delta\nu$ (μm^{-1}) is the wavenumber difference between T_1 and T_2 , the calculated k_{IC} for **TPA1** is $3.95 \times 10^{-10} s^{-1}$, which indicates that triplet excitons can only emit as photons from T_1 . To shed light on the origin of dual phosphorescence, we further conducted relative Gibbs free energies and the three global/local-minima geometries (T1-1 to T1-3) of T_1 -state **TPA1** (Figure 6). As Figure 6 shows, there are different **TPA1** conformations showing different T_1 states, which exhibit T_1 emissions contributed by donor and acceptor. Due to the existence of energy barriers (TS1 and TS2), the conformation transformation should be decided by transition energy, which can be compensated at room temperature while inhibited at low temperature, such as 77 K. T_1^H and T_1^L are not completely localized on donor and acceptor of **TPA1** perhaps because cyanobenzene is a type of extremely weak emitter, which is consistent with spectroscopic evidence that **TPA1** T_1^L emission is always dominant. Therefore, we further conducted the calculations of **TPA4**, where benzophenone is a better emitter than cyanobenzene. As anticipated, two T_1 species are localized on **TPA** and benzophenone subunits, perfectly corresponding with the experimental results (Figure 5c and S14). Therefore, we can conclude that the dual phosphorescence mainly originates from T_1 states of **TPA** derivatives in specific conformations, lining up with the dual-phosphorescence model in Figure 4f.”

Figure 6. Theoretical computations on the conformations and molecular orbital transitions for dual phosphorescence. Calculated relative Gibbs free energies and

global/local-minima geometries (T1-1, T1-2, T1-3 and T1-4) of T₁-state, the related transition states (TS) between these geometries, the optimal geometries with dihedrals of highlighted atoms, and main molecular orbital transition contributions for T₁ emissions (>10%) (H= HOMO, L=LUMO).

To test the generality whether highly twisted organic phosphors are prone to exhibit temperature-dependent multi-emissive phosphorescence, we synthesized other structurally unrelated compounds and found that the hypothesis works very well. The results will be published in the near future.

4. Researches using anti-Kasha's rule to obtain and demonstrate dual phosphorescence have been reported. In 2017, Tang group (Nat. Commun. 2017, 8, 416) reported the dual RTP phenomenon which arise from the low- and high-lying triplet states.

Our reply: Thanks for this piece of information. For Prof. Tang's work, the dual phosphorescence is mainly from T₂ with sub-ms decay and T₁ with hundreds of ms lifetime, of which the mechanism is different from our study, as we have shown above in Reply to Comment 3. Indeed, in order to achieve more than one emitting states from same electronic spin, examples of the anti-Kasha's rule are quite rare. However, our method of using sp³-linked highest twisted donor/acceptor can become quite general and is likely to spur interests among chemists and material scientists wishing to come up with molecules of novel luminescence properties.

5. In Figure 1, authors should provide the full name of EWG.

Our reply: We greatly appreciate pointing out this question. The full name of EWG (electron-withdrawing group) has been provided in Figure 1.

6. We suggest the author to use X-ray diffraction spectra to investigate the microstructure of the aggregates in THF/H₂O mixtures.

Our reply: We cordially appreciate giving us this valuable suggestion. However, we were told by our Hefei National Synchrotron Facility (operated by our university) staff that the concentration at which these aggregates are generated in THF/H₂O mixtures is insufficient to merit a reliable experimental result (~10⁻³ mol/L). However, credible alternative, scanning electron microscopy (SEM) measurement, was advised to us. We have presented these SEM images in the supporting information (Supplementary Figure 4) and their related discussions are highlighted in the manuscript. In short, SEM images (Supplementary Figure 4) show that some of these aggregates are in fact ordered and form nanocrystals (**TPA1**, **TPA2** and **TPA5**); the increased particle sizes of **TPA3** and **TPA4** may originate from reduced intermolecular interactions given that they exhibit high molecular complexity.

Supplementary Figure 4. Scanning electron microscopy images of **TPA1-6**.

7. On Page 9 of the manuscript, line NO.139, the authors mentioned that “an sp^2 counterpart and thus less electronically coupled at low temperature when vibrations are inhibited” The authors should provide the photophysical data of counterpart molecules.

Our reply: We thank the reviewer for suggesting such an experimental – we actually had a parallel project going on (which is less exciting) with studies on many of the sp^2 counterparts. The sp^2 version has a weak RTP band with a tiny shoulder band, matching fluorescence emission, that has the same decay kinetics attributed to delayed fluorescence. The data will be published elsewhere by a different author. However, a sneak peek is available below (it seems that the sp^2 linkage renders the RTP state localized on the diphenyl moiety instead of the **TPA**; this is understandable since the diphenyl $^3\pi\pi^*$ has the lowest triplet excited state). The conclusion was accidentally “leaked” without showing evidence by a co-author from the current manuscript, who is the lead author of the parallel study. We have deleted the comment regarding comparisons to the sp^2 counterpart, which was unnecessary anyway.

Responses to the Comments and Suggestions of Reviewer 2

General Comment: In this manuscript, Zhang et al. reported that donor- sp^3 linker-acceptor structures can generate a fascinating dual phosphorescence, which is proved tunable via modulation of temperature as a result of strong or weak electronic

coupling. Furthermore, through rationally decrease the electron-withdrawing ability of acceptor group, the authors made the visual phosphorescence color change more dramatic. This finding offered a vital clue on modulating complex excited-state dynamics in molecular systems, indicating a universal principle for designing dual-phosphorescence thermochromic materials. The elaborate experimental data and theoretical calculations signify the excellent quality of this work. Considering its novelty and foreseeable wide-ranging influence in many related fields, therefore, I strongly recommend its publication in Nature Communications, after the authors address the following concerns.

Our reply: We greatly appreciate the reviewer's recognition on novelty and importance of our work. As the reviewer stated, this finding offered a vital clue on modulating complex excited-state dynamics in molecular systems, and we believe the field will be excited to see our report. Our point-to-point response is given below.

1. Figure 1 shows that all five TPA compounds exhibit local excitation, while the authors stated that the S1 state is a dark state with the CT character. If so, why do the emission spectra show CT emission character? The authors should give more explanation.

Our reply: we greatly thank the reviewer for raising the valuable question. According to the calculations (Figure 2 and S1), we concluded that the absorptions and S1 are dominated by local excitations (LEs) and dark CT states, respectively. It explains why the TPAs in the isolated state in solution are non-emissive due to the excitation energy dissipation via the intramolecular rotations of TPA moiety and the nonradiative transition of the dark CT states. When the aggregates gradually form as increasing the fraction of water, the emissive S₁ state contributed by the local excitation of TPA appears. The emissions of the aggregates are the combination of the CT and LE states, which is also demonstrated by the calculation in Figure 6. Therefore, in Figure 2b, the emissions show a gradual blue shift as increasing the ratio of water due to the enhanced contribution of LE states of TPA.

2. From Figure 6, all transitions for T1 and T2 include the CT character. How do the authors confirm the radiative decays are from donor and acceptor, respectively?

Our reply: Thanks for pointing out this question. In previous Figure 6a and 6b, which have been moved, the transition configurations of T₁ and T₂ are obtained via the vertical excitations, not emission processes. Here, we are sorry to cause the reviewer this confusion. To elucidate the mechanism of dual phosphorescence, we further calculated relative Gibbs free energies (in kcal/mol) and the three global/local-minima geometries (T1-1 to T1-3) of T1-state of TPA1 (Figure 6). We found that despite the relatively large energy gap ($\Delta E_{T1-T2} > 0.55$ eV) the internal conversion between T₂ and T₁ is in the picosecond range. However, it has to be noted that there are some transition barriers between TPA1 conformations, of which at room temperature the

conversions among different conformations become more possible via TS2 and TS1 barriers while at lower temperature, such as 77 K, the conversions should be inhibited. Furthermore, a stronger piece of evidence is shown in Figure 5, where the two RTP bands with characteristic spectroscopic features belonging to donor and acceptor, respectively, are revealed when the temperature is decreased below 150 K. This experimental evidence is almost irrefutable. Therefore, we can conclude that the dual phosphorescence mainly originates from the T₁ state of **TPA** derivatives in specific conformations. The relevant descriptions have been presented in the computational investigation section and highlighted in yellow.

3. In caption of Figure 6, a letter “d” must be lost.

Our reply: Thanks for the reviewer’s careful inspection. “d” has been added as the form of (d) after **TPA2**.

4. For TPA6, it is interesting that the T2 band increases more obviously than T1 with temperature decreasing. I suggest the authors to give some detailed explanations.

Our reply: we cordially appreciate this important question. As we have stated in the Response, a new, more persuasive interpretation is provided (see Reply to Comment 3 of Reviewer 1). In short, we find that when there are more than one emitting states (local and global minima) along the T₁ potential energy surface, the ISC process which takes place at a higher electronic potential energy surface, will become temperature dependent. The reason is obvious – higher kinetic energy results in a larger shift to the right side of the equilibrium position, so that it is more favourable to go through ISC to relax to the lower emitting state. At lower temperatures, the ISC takes place closer to the equilibrium position, so that the higher emitting state becomes more prevalent. We hope to set up our own transient Raman system to substantiate the explanation soon.

5. According to UK English, “behavior” and “color” are encouraged to be replaced as “behaviour” and “colour”, respectively.

Our reply: Thanks for reviewer’s valuable suggestion. We have revised the related words.

Responses to the Comments and Suggestions of Reviewer 3

General comment: The manuscript by Zhang et al. described a series of purely organic AIE-active luminophores with efficient dual phosphorescence. By utilizing an sp³ linker as the spacer between the donor and acceptor to break the spatial overlap between HOMO and LUMO, efficient ISC process can be realized. The photophysics were fully understood by DFT calculations and temperature-dependent PL analysis. This work shows “donor-sp³ linker-acceptor” should have the potential to become a

new generation of purely organic RTP building block and platform, which will be inspirational and of great significance for the RTP community. Therefore, I would like to recommend its timely publication in Nat. Commun. after the following minor revisions.

Our reply: Thanks for reviewer's positive comments. This design principle could bring the RTP community a new strategy for developing versatile AIE-active RTP emitters. As the reviewer pointed out, this work has the "potential to become a new generation of purely organic RTP building block and platform." Our point-to-point response is presented as follows.

1. As shown in the MS, all these compounds consist of donor and acceptor subunits. Figure 1a shows the major UV absorption peaks around 300 nm, mainly ascribed to local excitation of TPA. Are these peaks influenced by solvents? The UV-vis absorption spectra in different solvents should be collected.

Our reply: Thanks for raising this important question. The UV-vis absorption spectra have been collected (presented in Supplementary Figure 2). As the solvent polarity changes, there is almost no absorption alternation for the main bands of **TPA1-6**, indicating CT states are dark state in the solution state, which is also backed by the calculation in Supplementary Figure 1.

Supplementary Figure 2. UV-vis absorption spectra of (a) TPA1, (b) TPA2, (c) TPA3, (d) TPA4, (e) TPA5, and (f) TPA6 in various optically dilute solvents.

2. For the dual phosphorescence phenomenon, the author assigned them as emissions from the donor and acceptor, respectively. How does the temperature influence the energy transfer process between the two different triplet states?

Our reply: we thank the reviewer for pointing out this question. To further shed light on the origin of dual phosphorescence, which is also commented by Reviewer 1 (Comment 3) and Reviewer 2 (Comment 3). As we have stated in our revision, the most likely mechanism is conformation-dependent emitting states. Our binary mixture experiment show that energy transfer is existential but perhaps trivial in that distinct dual RTP bands are observed in the mixed crystal. The revisions are provided in yellow highlights, which should be very convincing at this stage, and we welcome any new insight from the reviewer regarding the interesting phenomenon.

3. To better understand the origin of dual phosphorescence, NTO transition calculations should be carried out.

Our reply: we greatly thank the reviewers for giving this valuable suggestion. Similar to Comment 3 from Reviewer 1 and Comment 2 from Reviewer 2, we have taken TPA1 as an example to conduct relative Gibbs free energies (in kcal/mol) and the three global/local-minima geometries (T1-1 to T1-3) of T₁-state, and the transition states between these geometries (TS1 and TS2). The calculation results show that the dual phosphorescence originates from T₁ states in different conformations.

4. Given that in the aggregated state for TPA2-5 there is almost no RTP from a higher triplet state, do intermolecular interactions significantly influence the dual phosphorescence?

Our reply: Thanks for raising this question. Again, based on our spectroscopic data, we did see one of the two emitting states being a dominant one due to aggregation. Yes, we believe that such a process does exist in the solid.

5. For Figure 5 caption, “showing two triplet-state emissions” should be added for a better understanding of these spectra.

Our reply: we cordially appreciate the reviewer’s suggestion. The related revision has been done.

6. A minor typo: in figure 7, the digits of I455/425 should be subscript.

Our reply: Thanks for pointing out this typo. We have amended it.

REVIEWERS' COMMENTS

Reviewer #1 (Remarks to the Author):

The revision could be accepted since all the raised concerns have been well addressed.

Reviewer #2 (Remarks to the Author):

In the revised version of the manuscript, the authors have addressed all issues. Therefore, I strongly recommend its publication in Nature Communications as it is.

Reviewer #3 (Remarks to the Author):

The authors have adequately revised their manuscript according to my previous comments and suggestions. The quality of the manuscript has been improved after the revision. I do not have further criticism of the work. The revised manuscript can be published in its present form.

Replies to Reviewers' Comments for Manuscript NCOMMS-20-32351A

Responses to the Comments and Suggestions of Reviewer 1

Comment: The revision could be accepted since all the raised concerns have been well addressed.

Our reply: We greatly appreciate the acceptance and favorable comment from Reviewer 1. Thanks to your comments, the quality of this manuscript has been improved.

Responses to the Comments and Suggestions of Reviewer 2

Comment: In the revised version of the manuscript, the authors have addressed all issues. Therefore, I strongly recommend its publication in Nature Communications as it it.

Our reply: We cordially thank the reviewer for positive comments and valuable advice, which have significantly improved our research.

Responses to the Comments and Suggestions of Reviewer 2

Comment: The authors have adequately revised their manuscript according to my previous comments and suggestions. The quality of the manuscript has been improved after the revision. I do not have further criticism of the work. The revised manuscript can be published in its present form.

Our reply: We cordially appreciate these valuable suggestions for improving our research. Thanks for acceptance of our research.